# Plant mobile domain proteins ensure Microrchidia 1 expression to fulfill transposon silencing

Lucas Jarry[1,2], Julie Descombin[1,2,*], Melody Nicolau[1,2,*], Ange Dussutour[1,2], Nathalie Picault[1,2], Guillaume Moissiard[1,2]

Silencing of transposable elements (TEs) is an essential process to maintain genomic integrity within the cell. In *Arabidopsis*, together with canonical epigenetic pathways such as DNA methylation and modifications of histone tails, the plant mobile domain (PMD) proteins MAINTENANCE OF MERISTEMS (MAIN) and MAIN-LIKE 1 (MAIL1) are involved in TE silencing. In addition, the MICRORCHIDIA (MORC) ATPases, including MORC1, are important cellular factors repressing TEs. Here, we describe the genetic interaction and connection between the PMD and MORC pathways by showing that *MORC1* expression is impaired in *main* and *mail1* mutants. Transcriptomic analyses of higher order mutant plants combining *pmd* and *morc1* mutations, and *pmd* mutants in which *MORC1* expression is restored, show that the silencing defects of a subset of TEs in *pmd* mutants are most likely the consequence of *MORC1* down-regulation. Besides, a significant fraction of up-regulated TEs in *pmd* mutants are not targeted by the MORC1 pathway.

## Introduction

Transposable elements (TEs) are highly repeated, self-replicating genetic elements that are capable of invading the host genome through the process of transposition (1). TEs are predominantly enriched in pericentromeric constitutive heterochromatin, although they can also occupy chromosome arms (2). When occurring within a gene, TE transposition can disrupt gene sequence and function with dramatic consequences for the host cell. Thus, to maintain its genome integrity, the cell has elaborated several epigenetic pathways, such as DNA methylation and histone modifications that repress TEs (3, 4). In plants such as *Arabidopsis thaliana*, DNA methylation occurs in three different cytosine contexts that are mCG, mCHG, and mCHH (where H is A, T, or C), involving specialized DNA methyltransferases (5). DOMAINS REARRANGED METHYLTRANSFERASE 2 (DRM2) is required for de novo DNA methylation in all sequence contexts through the RNA-directed

DNA methylation (RdDM) pathway and in the maintenance of mCHH. METHYLTRANSFERASE 1 (MET1) is essential for the maintenance of virtually all mCG, whereas CHROMOMETHYLASE 2 (CMT2) and CMT3 are involved in mCHG maintenance. CMT2 can also mediate mCHH maintenance at specific genomic locations (6, 7). Besides DNA methylation and histone modifications, several epigenetic factors cooperate to repress TEs. These sophisticated epigenetic pathways converge toward TEs to maintain them silenced, acting either synergistically or redundantly (1). MICRORCHIDIA (MORC) proteins are ATPases conserved in most eukaryotes, playing a major role in TE and gene silencing in plants, nematodes, and mammals (8, 9). In *A. thaliana*, MORC1 physically interacts with MORC6 and with MORC4, MORC7, and RdDM factors to maintain heterochromatic TEs condensed (10, 11). It has been proposed that MORC proteins would repress TEs using a DNA loop-trapping mechanism to compact chromatin (12). Another pathway involves MAINTENANCE OF MER-ISTEMS (MAIN) and MAIN-LIKE 1 (MAIL1) that are two plant mobile domain (PMD) proteins, originally identified as essential factors for plant development and genome integrity (13, 14). MAIN and MAIL1 physically interact together, forming a molecular complex with the presumably inactive serine/threonine phosphoprotein phospha-tase (PPP) called PP7-LIKE (PP7L). The three proteins are required for TE silencing and the proper expression of a common subset of genes (15, 16, 17). Synergistic effects were described between MAIN, DRM2, and CMT3 pathways (16). Nevertheless, the mode of action of PMD proteins remains largely unclear, and their involvement in TE silencing is elusive.

In this study, we report the complex interplay between the PMD and MORC1 pathways. We show that *MORC1* belongs to the genes that are commonly down-regulated in several single- and higher order *pmd* mutants. Based on these observations and considering the major role of *MORC1* in TE silencing, we hypothesized that *MORC1* down-regulation could at least partially explain the TE si-lencing defects observed in the *pmd* mutants. To address this question, we decided to undertake two approaches: first, to deci-pher the genetic interaction between the PMD and MORC1 pathways by analyzing misregulation of TE and gene expression in *main morc1* and *mail1 morc1* double mutants; and second, to use a transgene-based approach to rescue *MORC1* expression in *pmd*

---

[1]LGDP-UMR5096, CNRS, Perpignan, France [2]LGDP-UMR5096, Université de Perpignan Via Domitia, France

Correspondence: guillaume.moissiard@univ-perp.fr
Melody Nicolau's present address is IAB Centre de Recherche UGA/Inserm U 1209/CNRS UMR 5309 Site Santé, La Tronche, France
*Julie Descombin and Melody Nicolau contributed equally to this work

---

 

# A

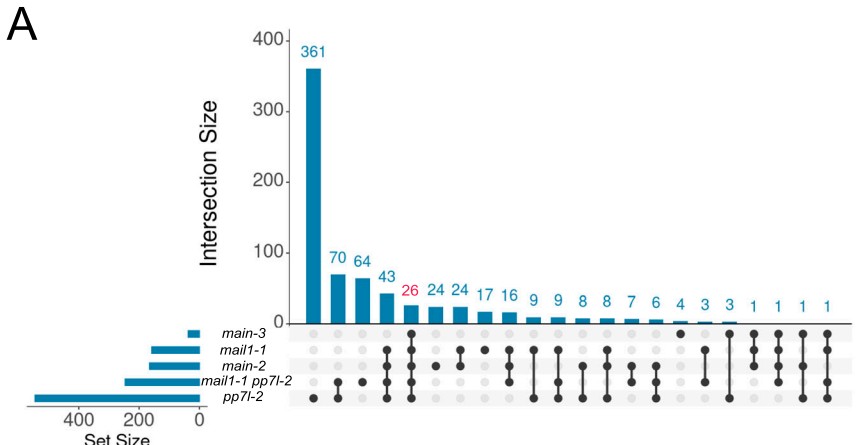

# B

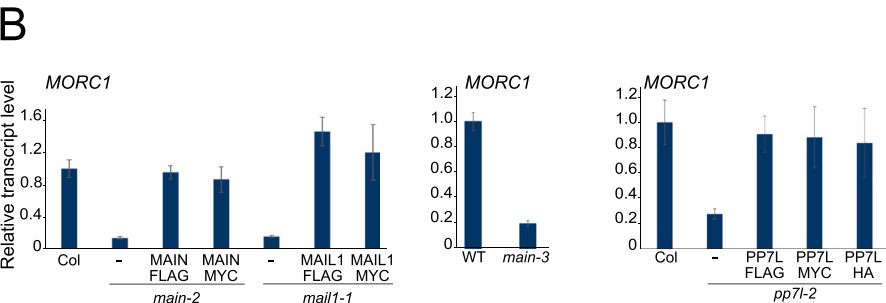

**Figure 1. MORC1 is down-regulated in *pmd* and *pp7l* mutants.**
**(A)** UpSet plot analyses allowing to visualize in a matrix layout the intersections of down-regulated gene datasets in the *main-3* hypomorphic mutant, *main-2*, *mail1-1*, *pp7l-2* single-null, and *mail1-1 pp7l-2* double-null mutants as described in reference 16.
**(B)** Relative expression of *MORC1* mRNA levels assayed by reverse transcription coupled to quantitative PCR (RT–qPCR) in corresponding *pmd* and *pp7l* mutants and complementing lines. RT–qPCR analyses were normalized using the housekeeping *RHIP1* gene, and transcript levels are represented relative to WT Columbia (Col) or *ATCOPIA28::GFP* in WT controls (16). Error bars indicate SD based on three independent biological replicates.
Source data are available for this figure.

mutants, which demonstrates that silencing of a fraction of up-regulated TEs in *main* and *mail1* mutants can be complemented by supplying MORC1 in *trans*.

# Results

### *MORC1* is down-regulated in *pmd* mutants

By surveying the genes that were misregulated in the *main-3* hypomorphic mutant, in the *main-2 mail1-1* and *pp7l-2* null mutants (hereafter called *main*, *mail1*, and *pp7l* in the text), and in higher order mutants thereof, we identified 26 genes that were commonly down-regulated in all the genetic backgrounds (Fig 1A and Table S1) (16). 25 of them carry a DNA motif in their promoter that was previously named the "DOWN" motif (16). Although we could not define any enrichment of gene ontology (GO) term among these genes, we found out that *MORC1*, which carries the "DOWN" motif in its promoter, belonged to the list of down-regulated loci (Table S1). This was further confirmed by reverse transcription coupled to quantitative PCR (RT–qPCR) experiments showing a fivefold decrease in all tested mutants in comparison with WT Columbia (Col) control (Fig 1B). Furthermore, *MORC1* expression could be rescued in *main*, *mail1*, and *pp7l* null mutants that were complemented with the respective epitope-tagged genomic *PMD* or *PP7L* constructs (Fig 1B). Thus, the two PMD MAIN and MAIL1 proteins, and their interactor PP7L, are required for the proper expression of *MORC1*, and to some extent, *pmd* and *pp7l* mutants can be seen as *morc1* knocked-down mutants.

### The *pmd morc1* double-null mutants do not exacerbate TE silencing defects

To evaluate the effect of *MORC1* down-regulation on TE activation observed in the *pmd* mutants, we decided to analyze the genetic interaction between PMD and MORC1 pathways by creating *main morc1* and *mail1 morc1* double-null mutants using the *morc1-2* null allele (hereafter called *morc1*). Although *morc1* mutant and WT Col plants are undistinguishable, *main-2* and *mail1* single-null mutants display a strong developmental phenotype that is not exacerbated by introducing the *morc1* null mutant allele (Figs 2A and S1A). We then performed RNA-sequencing (RNA-seq) analyses using *main*, *morc1* single-null, and *main morc1* double-null mutants, and evaluated TE and gene misregulation in comparison with WT control plants (Fig 2B–D and Table S2). Principal component analyses showed that biological replicates of each genetic background clustered together, and remarkably, replicates of *main* and *main morc1* mutants tend to group together (Fig S1B). Although up-regulated TEs were mostly pericentromeric, misregulated genes spanned the whole five chromosomes (Fig S1C). Furthermore, for up-regulated TEs and genes, comparative analyses identified significant numbers of loci that were commonly misregulated in the three mutant backgrounds (Fig 2E). We noticed that several TEs were apparently up-regulated only in the *main* single but not in the *main morc1* double mutant or vice versa (Fig 2B and E). However, by analyzing more precisely the expression level of these TEs in each mutant background, we observed that overall, they seemed to be similarly up-regulated in *main* and *main morc1* mutants, but, most likely, did not pass our stringent RNA-seq threshold ($\log_2 \geq 2$ or $\log_2$

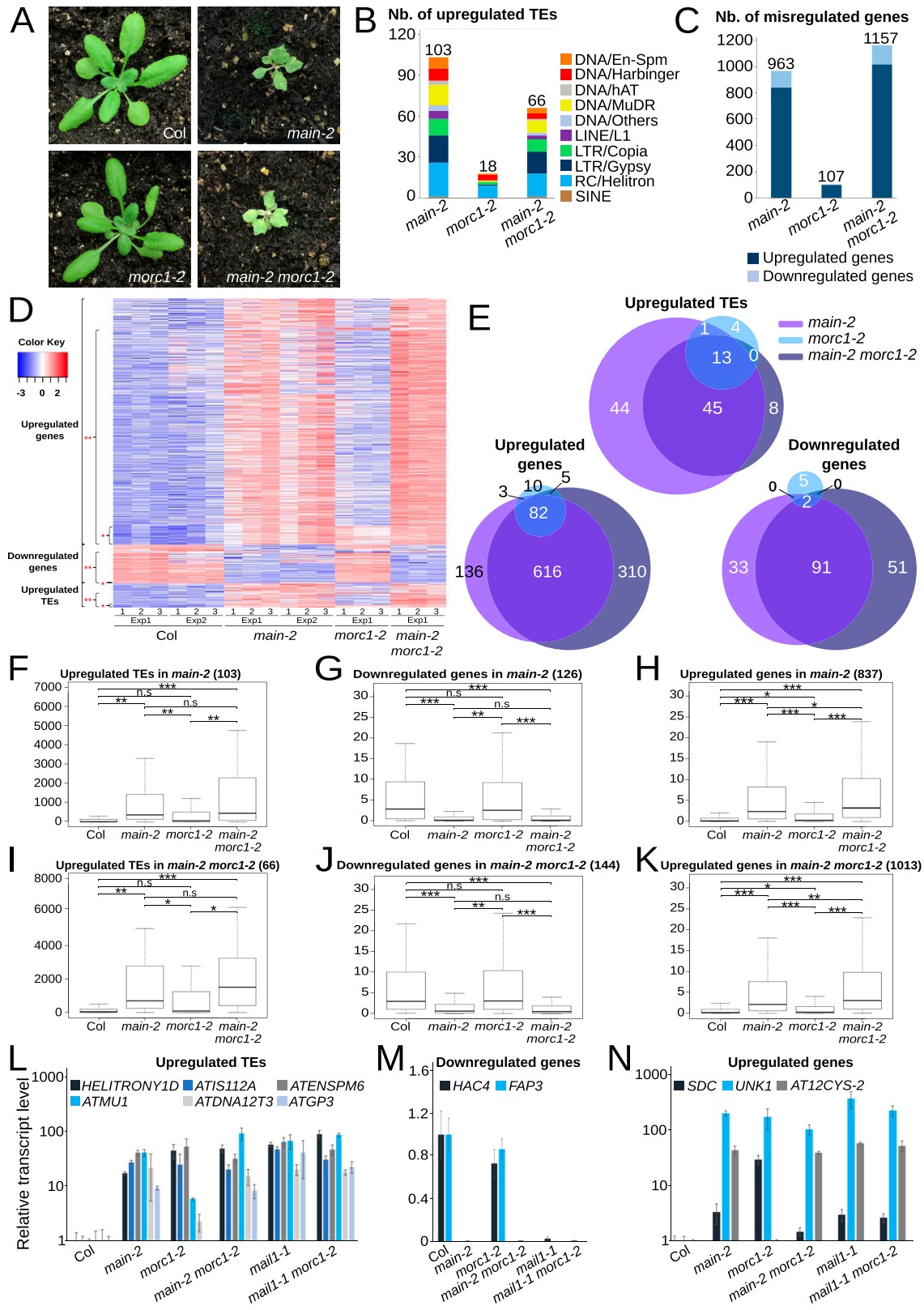

**Figure 2.   TE silencing defects are not aggravated by combining *pmd* and *morc1* mutations.**
**(A)** Representative pictures of 3-wk-old *main-2*, *morc1-2* single, and *main-2 morc1-2* double mutants in comparison with WT Col plant. **(B)** Number of up-regulated TEs in *main-2*, *morc1-2*, and *main-2 morc1-2*, classified by the TE superfamily. **(C)** Number of misregulated genes in *main-2*, *morc1-2*, and *main-2 morc1-2*. **(D)** Heatmap showing misregulated loci in several biological replicates of *main-2*, *morc1-2*, and *main-2 morc1-2* in comparison with WT Col. * represents loci that are commonly misregulated in

≤ −2 fold change, adjusted $P$ < 0.01; Fig 2D). To statistically validate this hypothesis, we performed boxplot analyses using TEs that were up-regulated in *main* or *main morc1* mutants, which confirmed that there was no significant difference between these two genetic backgrounds regarding the extent of TE up-regulation (Fig 2F and I). In contrast, the *morc1* null mutant showed a milder up-regulation of TEs than the *main* mutant, in which *MORC1* is knocked down, suggesting that MAIN plays a broader role in TE silencing than MORC1 (Fig 2B, F, and I). Similar analyses with genes that were down-regulated in *main* or *main morc1* mutants showed comparable results, with no significant difference between the two genetic backgrounds (Fig 2G and J). However, for up-regulated genes in *main* or *main morc1* mutants, we observed significant differences between the *main* single and *main morc1* double mutants, indicating a possible synergistic effect of the two mutations at these genomic locations (Fig 2H and K). This could also be explained as a consequence of another down-regulated gene deriving from the *main* background. Search for GO term enrichment revealed that up-regulated genes were significantly associated with the term "response to stress," whereas down-regulated genes in *main* and *main morc1* mutants were related to "response to red light" and "oxidoreductase activity" terms (Fig 2E and Table S3).

These observations were further confirmed at several TEs and misregulated genes by performing RT–qPCR experiments, which also included *mail1* and *mail1 morc1* mutant backgrounds (Fig 2L–N). Altogether, these analyses revealed that *pmd* mutants showed a wilder up-regulation of TEs and misregulation of genes in comparison with the *morc1* mutant. However, cumulating the *pmd* and *morc1* mutations did not significantly aggravate the TE silencing defects observed in the *main* single mutant.

### The *pUBQ-MORC1* construct complements the TE silencing defects of *morc1* null mutant

We showed that *MORC1* is down-regulated in the *pmd* mutants, the *pmd* and *morc1* null mutants share a common subset of up-regulated TEs and genes, and there is no significant difference in TE silencing defects between *main* single and *main morc1* double mutants (Figs 1 and 2). This suggests that *MAIN* and *MAIL1* are epistatic to *MORC1*, and *MORC1* down-regulation might contribute, at least partially, to the TE silencing defects observed in the *pmd* mutant. To test this hypothesis, we engineered the *pUBQ-MORC1* construct, in which the *MORC1* coding sequence fused to a 3xFLAG epitope was cloned under the control of the housekeeping gene *UBIQUITIN10* promoter (*pUBQ*) (Fig 3A) (18). The rationale was that placing *MORC1* under *pUBQ* control would efficiently restore *MORC1* expression because *UBQ10* transcription is not impaired in *main* and *mail1* mutants as seen in RNA-seq data (Fig S2A). *pUBQ-MORC1*

was thus introduced in *main* and *mail1* mutants by plant transformation to generate *pUBQ-MORC1/main* line 1 and line 2, and *pUBQ-MORC1/mail1* line 1 and line 2. To assay *pUBQ-MORC1* functionality, the transgenes deriving from *pUBQ-MORC1/main* line 2 and *pUBQ-MORC1/mail1* line 1 were introduced into the *morc1* null mutant by crosses to generate *pUBQ-MORC1/morc1* line 1 and *pUBQ-MORC1/morc1* line 2, respectively. The accumulation of the MORC1-FLAG protein in each line was confirmed by Western blots, and RT–qPCR experiments demonstrated that the accumulation of MORC1-FLAG in both lines was sufficient to restore the silencing of several misregulated TEs and DNA-methylated genes in *morc1-2* (Fig 3B and C). Thus, *pUBQ-MORC1*–derived MORC1-FLAG is a functional protein.

### *pUBQ-MORC1* expression is sufficient to rescue the silencing at a subset of TEs in *pmd* mutants

To assess the effect of the functional MORC1-FLAG protein in the *pmd* mutants, we analyzed the four *pUBQ-MORC1/main* and *pUBQ-MORC1/mail1* lines. As expected, the developmental phenotype of *main* and *mail1* mutants was not complemented in *pUBQ-MORC1/main* and *pUBQ-MORC1/mail1* lines (Fig 3D). *pUBQ-MORC1* expression in the four lines was checked at the RNA and protein levels, confirming the accumulation of the MORC1-FLAG protein (Figs 3E and S2B). We then investigated the capacity of MORC1-FLAG to rescue the silencing defects of several TEs and DNA-methylated genes by RT–qPCR experiments. Remarkably, the four *main* and *mail1* mutant lines expressing the *pUBQ-MORC1* transgene showed a significant reduction in the expression of TEs and DNA-methylated genes in comparison with respective control mutant backgrounds (Fig 3F and G). Furthermore, at loci such as *ATREP18* or *UNK1*, the strength of silencing was back to the WT level (Fig 3F and G). Conversely, in two additional independent lines called *pUBQ-MORC1/main* negative (neg.) and *pUBQ-MORC1/mail1* neg. that did not accumulate the MORC1-FLAG protein, TE and DNA-methylated gene silencing was not rescued, with expression levels similar to the *main* and *mail1* mutant controls (Fig S2C and D).

To fully evaluate the effect of rescuing *MORC1* expression on TE silencing in the *pmd* mutants, we decided to extend our analyses by performing RNA-seq using the four *pUBQ-MORC1/pmd* lines accumulating the MORC1 protein. In each line, we could identify several complemented TEs, that is, TEs that were repressed in *pUBQ-MORC1/pmd* lines while identified as up-regulated in the respective *pmd* mutants (Fig 4A and B and Tables S2, S4, and S5). Although we could observe variation between independent lines, they shared significant fractions of complemented TEs, all of them being pericentromeric (Fig 4B and Table S5). As expected, boxplot analyses did not show any differences in TE up-regulation between *main* and *mail1* (Fig 4C). Although rescuing *MORC1* expression in the

the three mutant backgrounds. ** represents loci that are misregulated in *main-2 morc1-2*. **(E)** Venn diagram analyses representing the overlaps between misregulated loci in *main-2*, *morc1-2*, and *main-2 morc1-2*. Fisher's exact test statistically confirmed the significance of overlaps ($P$ < 10$^{-3}$). **(F, G, H)** Boxplot analyses between *main-2*, *morc1-2*, and *main-2 morc1-2* mutants in comparison with WT Col showing average RPKM values of up-regulated TEs (F), up-regulated genes (G), and down-regulated genes (H) in *main-2*. **(I, J, K)** Same as (F, G, H) using misregulated loci in *main-2 morc1-2* as defined by ** in panel (D). $P$-values were calculated using a Wilcoxon test; n.s, not significant; *$P$ < 0.05; **$P$ < 10$^{-6}$; and ***$P$ < 10$^{-12}$. **(L, M, N)** Relative expression analyses of up-regulated TEs, down-regulated genes, and up-regulated genes in the different genotypes assayed by RT–qPCR. RT–qPCR analyses were normalized using the housekeeping *RHIP1* gene, and transcript levels in the different mutants are represented relative to WT Col. Error bars indicate SD based on three independent biological replicates.
Source data are available for this figure.

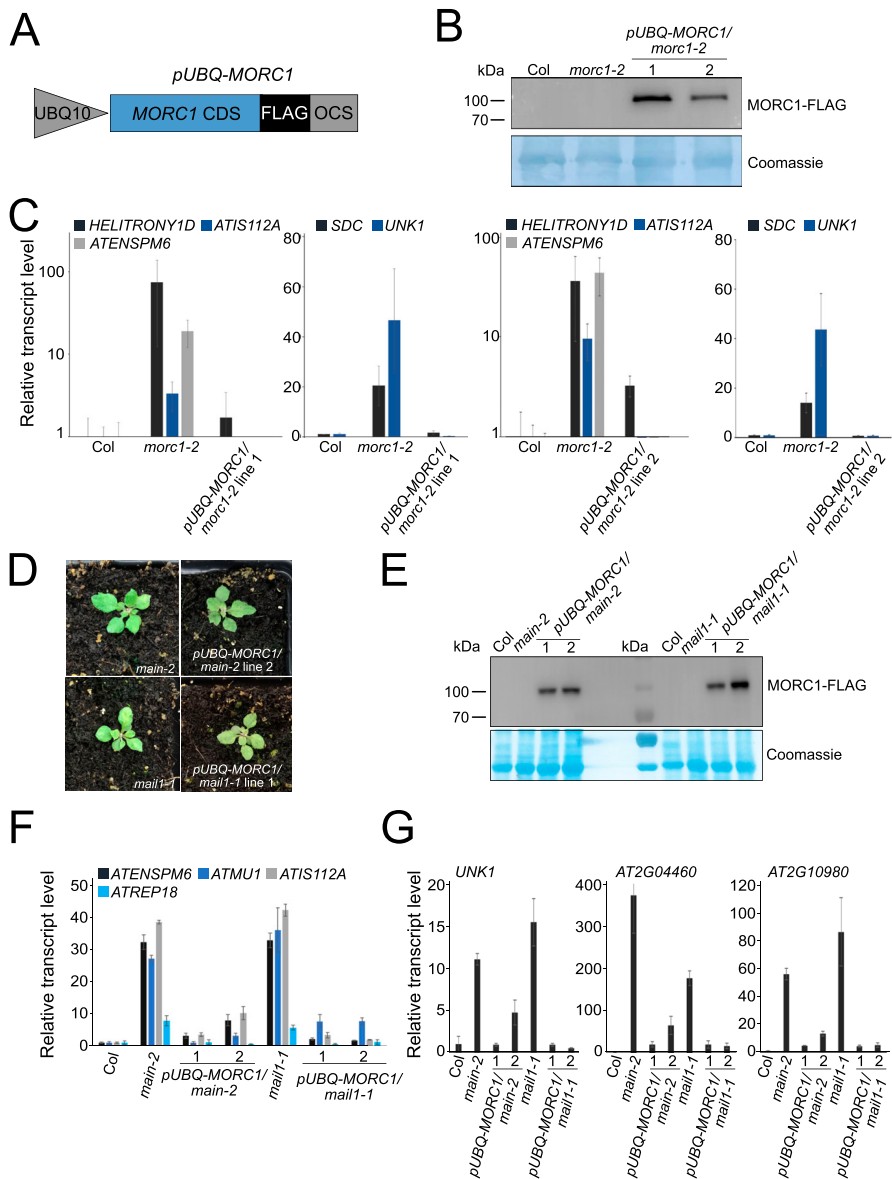

**Figure 3. *pUBQ-MORC1* transgene can complement the silencing defects of several TEs in *morc1*, *main*, and *mail1* mutants.**
**(A)** Schematic representation of the *pUBQ-MORC1* transgene. *MORC1 CDS-3xFLAG* is under the control of *UBQ10* promoter and *octopine synthase* terminator. **(B)** Western blots using anti-FLAG antibody showing the accumulation of FLAG-tagged MORC1 protein in two *pUBQ-* independent *MORC1/morc1-2* lines. WT Col and *morc1-2* plants are used as negative controls. Coomassie staining of the membrane is used as a loading control; kD, kilodalton. **(C)** Relative expression levels of up-regulated TEs and DNA-methylated genes in the two *pUBQ-MORC1/morc1-2* lines and *morc1-2* control plants assayed by RT–qPCR. RT–qPCR analyses were normalized using the housekeeping *RHIP1* gene, and transcript levels in the two genetic backgrounds are represented relative to WT Col. Error bars indicate SD based on three independent biological replicates. **(D)** Pictures of 3-wk-old *pUBQ-MORC1/main-2*, *pUBQ-MORC1/mail1-1*, and corresponding untransformed *pmd* mutants. **(E)** Same as (B) using two independent lines of *pUBQ-MORC1/main-2* and *pUBQ-MORC1/mail1-1* and WT Col, *main-2*, and *mail1-1* as controls. **(F, G)** Same as (C) using *pUBQ-MORC1/main-2* and *pUBQ-MORC1/mail1-1* lines in comparison with *main-2* and *mail1-1* mutants and relative to WT Col. Source data are available for this figure.

two *pmd* mutants did not fully restore TE silencing to the WT level, we could, however, observe that TE complementation was statistically significant for three of the four lines in comparison with their respective mutants (Fig 4D and E).

We next determined the fractions of misregulated genes in *main* and *mail1* mutants that were complemented by the *pUBQ-MORC1* transgene (Fig S3A and Tables S2, S4, and S5). Unlike TEs, we could only identify a handful of commonly complemented genes between independent *pUBQ-MORC1/main* and *pUBQ-MORC1/mail1* lines, with bigger variations between the lines (Fig S3B and C). Nevertheless, some of these lines showed complementation of misregulated genes that were statistically significant (Fig S3D–G). To explain the discrepancies between independent lines, we hypothesize that these variations are consequences of MORC1-unspecific effects occurring in each *pUBQ-MORC1* line. Furthermore, three stress response–related genes *LURP1*, *BG3*, and *WRKY38* identified

by RNA-seq as complemented in *pUBQ-MORC1/main* line 2 were not validated by RT–qPCR analyses, neither were *HAC4* and *FAP4* that are two genes previously identified as down-regulated in *main* and *mail1* (Fig S3H and I) (16). Importantly, up-regulated genes that were commonly complemented in the independent *pUBQ-MORC1/main* and *pUBQ-MORC1/mail1* lines are mostly DNA-methylated genes that are enriched in the pericentromeric regions with no GO term enrichment (Fig S3B and Tables S5 and S6). Among these genes, we found the DNA-methylated gene *UNK*, and the two transposable element genes *AT2G04460* and *AT2G10980*, validated by RT–qPCR (Fig 3G). Finally, we performed boxplot analyses using up-regulated TEs in *morc1*, which showed that for most of these TEs, the silencing was back to the WT level in the four *pUBQ-MORC1/pmd* lines (Figs 4F and S4A). Moreover, comparative analyses between commonly complemented TEs in *pUBQ-MORC1/main* or *pUBQ-MORC1/mail1* lines and up-regulated TEs in *pp7l* or *main-3* mutants showed

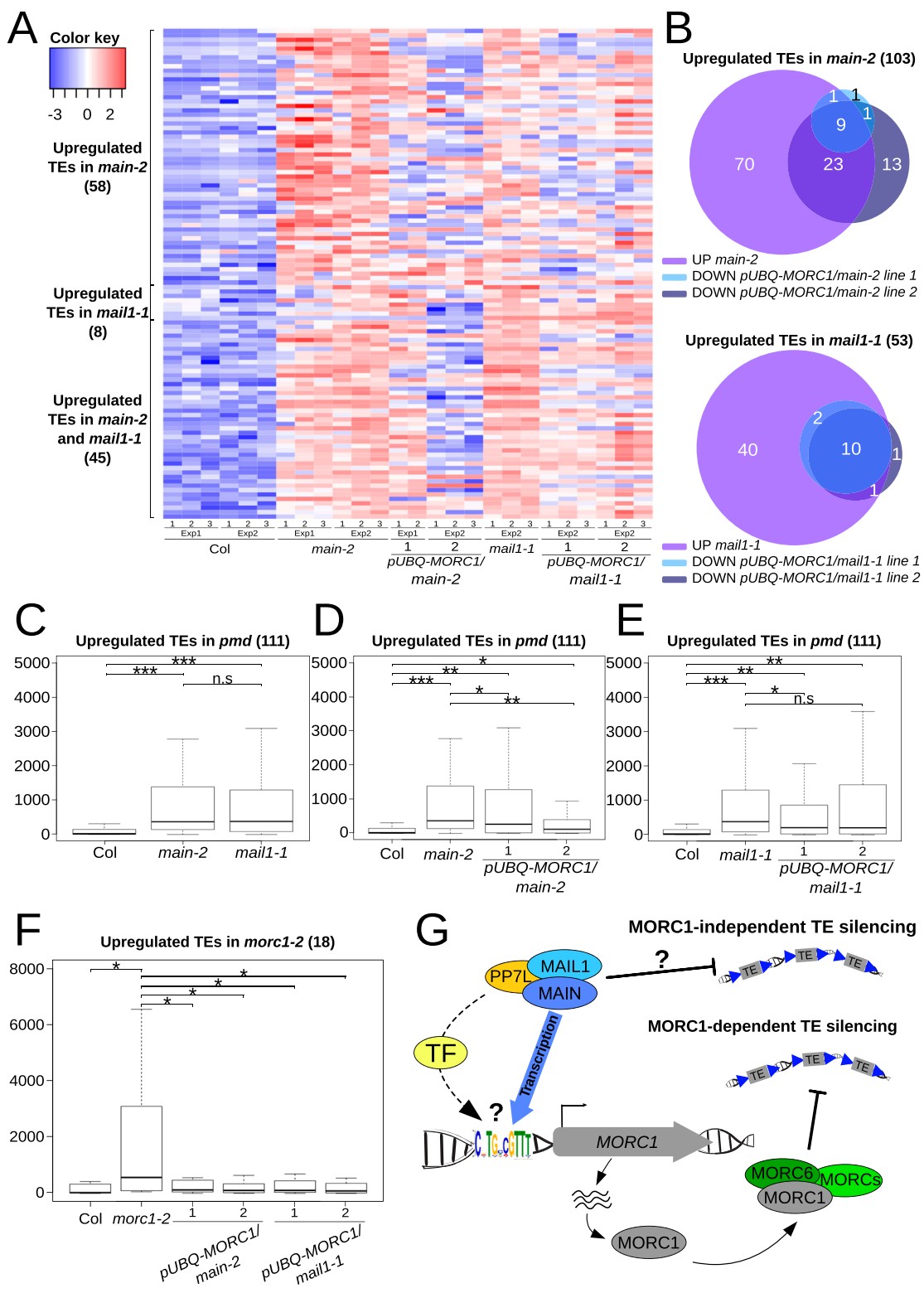

**Figure 4. Rescuing *MORC1* expression in *pmd* mutants efficiently restores the silencing of a fraction of TEs.**
**(A)** Heatmap representing up-regulated TEs in *main-2* and *mail1-1* mutants and their expression levels in four independent *pUBQ-MORC1/pmd* lines. **(B)** Venn diagram analyses showing the overlaps between down-regulated TEs in *pUBQ-MORC1/main-2* or *pUBQ-MORC1/mail1-1* lines over their respective mutant backgrounds and up-regulated TEs in *main-2* or *mail1-1* over WT Col. Fisher's exact test statistically confirmed the significance of overlaps ($P < 10^{-3}$). **(C)** Boxplot analyses between *main-2* and *mail1-1* showing average RPKM values of up-regulated TEs in *main-2* and *mail1-1* union in comparison with WT Col. **(D, E)** Boxplot analyses between two independent *pUBQ-MORC1/main-2* (D) or *pUBQ-MORC1/mail1-1* (E) lines and their respective *pmd* mutants showing average RPKM values of up-regulated TEs in *main-2* and *mail1-1*

significant overlaps (Fig S4B and C). Altogether, these results demonstrate that supplying MORC1 in *trans* in *pmd* mutants (i) efficiently restores the silencing of a fraction of TEs that are pericentromeric and up-regulated in *morc1* mutant; (ii) to a lesser extent rescues the expression of several up-regulated genes that are mostly repressed genes targeted by DNA methylation in WT; and (iii) finally has a minor effect on genes that are down-regulated in the *pmd* mutants.

# Discussion

### *MORC1* down-regulation in *pmd* mutants cannot explain their abnormal developmental phenotype

The PMD proteins MAIN and MAIL1 are involved in several aspects of plant development, and the massive misregulation of gene expression in *main* and *mail1* mutants is most likely accountable for their strong developmental phenotype (13, 14, 16). We showed here that *MORC1* is down-regulated in the *pmd* mutants and in *pp7l*, which is mutated for the MAIN and MAIL1 interactor, PP7L (Fig 1) (16, 17). By analyzing higher order combinations of *pmd* and *morc1* null mutations together with *pmd* plants in which *MORC1* expression is restored (*pUBQ-MORC1/pmd* lines), we conclude that *MORC1* down-regulation cannot account for the abnormal developmental phenotype of *pmd* mutants because *pmd* single and *pmd morc1* double mutants and *pUBQ-MORC1/pmd* lines are undistinguishable (Figs 2A and 3D). Moreover, the subset of genes whose expression is rescued in the *pUBQ-MORC1/pmd* lines are not the causal factors of the *pmd* developmental phenotype, and further work will be needed to address this question.

### Combining *pmd* and *morc1* null mutations does not exacerbate TE silencing defects of *pmd* mutants

Distinct epigenetic pathways cooperate to efficiently silence TEs, forming the so-called epigenetic "mille-feuille" (1, 19). Generally, cumulating mutations in different epigenetic pathways acting redundantly or synergistically leads to a dramatic aggravation of TE silencing defects. For instance, a synergistic effect was observed in plants combining the *morc6* and *morpheus' molecule 1 (mom1)* mutations (8). Similarly, introducing *drm2* and *cmt3* mutations into the hypomorphic *main-3* mutant allele led to a dramatic derepression of TEs (16). It was also shown for a handful of TEs that combining *mail1* and *morc6* mutations had a mild synergistic effect (15). Conversely, we observed that *main morc1* double mutant did not show genome-wide massive up-regulation of TEs in comparison with *main* (Fig 2B–E). Indeed, focusing on the subsets of up-regulated TEs in *main* or *main morc1* showed that there was no

significant difference in misregulation between *main morc1* and *main* mutants, which was further confirmed by RT–qPCR analyses including *mail1* and *mail1 morc1* mutants (Fig 2F, I, and L). However, TE silencing defects appeared stronger in *main* in comparison with the *morc1* mutant (Fig 2B, F, and I). We propose that the stronger effect of *morc6* mutation in comparison with *morc1* on TE derepression could explain the discrepancy observed between *mail1 morc1* and *mail1 morc6* (8, 10). Thus, combining *pmd* and *morc1* mutations does not exacerbate TE silencing defects, suggesting that the two pathways are connected, which is consistent with the fact that *MORC1* is down-regulated in *pmd* mutants.

### Rescuing *MORC1* expression in *pmd* mutants is sufficient to restore the silencing of a subset of TEs

To re-establish *MORC1* expression in *pmd* mutants, we introduced a FLAG-tagged MORC1 construct (*pUBQ-MORC1*) under the control of the *UBQ10* promoter (Fig 3A). We first showed that *pUBQ-MORC1* expression efficiently complemented the up-regulation of several TEs in the *morc1* null mutant, confirming that the protein is functional (Fig 3C). We then analyzed the effect of *pUBQ-MORC1* in *main* and *mail1* in four independent lines and found that the silencing of a subset of TEs was restored in this genetic material (Figs 3F and 4A–E). Remarkably, these TEs corresponded to a significant fraction of TEs that were also up-regulated in *morc1*, which is consistent with the fact that (i) *pmd* mutants can be seen as *morc1* knocked-down mutants and (ii) *main* and *main morc1* mutants display similar TE up-regulation phenotypes (Figs 4F and S4A). Based on these results, we propose a model in which the MAIN/MAIL1/PP7L complex is required for the proper expression of the MORC1 protein, which in turn ensures the silencing of a subset of TEs together with other MORC proteins, including MORC6 (Fig 4G). It is not known at the moment whether the MAIN/MAIL1/PP7L complex interacts with chromatin. This interaction could be direct or indirect through the recruitment of an unknown transcription factor that would recognize, for instance, the cis-regulatory DNA elements called "DOWN" motif that is enriched in the promoter of down-regulated genes—including *MORC1*—in *pmd* and *pp7l* mutants (Fig 4G and Table S1) (16). Another hypothesis would be that *MORC1* expression is regulated by a transcription factor acting downstream of the MAIN/MAIL1/PP7L complex.

Finally, this study revealed that a significant fraction of up-regulated TEs in the *pmd* mutants are not targeted by MORC1 (Fig 4G). One possibility is that the PMD proteins directly repress these TEs. A non-exclusive alternative would be that these TEs could also be targeted by an unknown factor that is impaired in the *pmd* mutants. Further studies will be essential to address these questions and to clarify the essential role of PMD proteins in TE silencing.

union in comparison with WT Col. **(F)** Same as (D, E) for up-regulated TEs in *morc1-2* as defined in Fig 2. **(C, D, E, F)** *P*-values of panels (C, D, E, F) were calculated using a Wilcoxon test; n.s, not significant; *$P < 0.05$; **$P < 10^{-6}$; and ***$P < 10^{-12}$. **(G)** In this model explaining the connection between the PMD and MORC1 pathways to repress TEs, *MORC1* transcription requires the MAIN/MAIL1/PP7L complex. This latter could either directly recognize the "DOWN" motif located within the *MORC1* promoter (CATGCAGTTT) or be recruited by an elusive transcription factor at this genomic location. Alternatively, *MORC1* expression would indirectly depend on the MAIN/MAIL1/PP7L complex through the action of a downstream transcription factor. Upon translation, the MORC1 protein associates with other MORC proteins to ensure efficient silencing of a subset of TEs (MORC1-dependent TE silencing). Importantly, the silencing of a significant fraction of TEs requires another pathway independent of MORC1 yet to be deciphered. This MORC1-independent TE silencing pathway could directly involve the PMD proteins or another factor regulated by the MAIN/MAIL1/PP7L complex. Source data are available for this figure.

# Materials and Methods

### Plant material and growing conditions

WT and mutant lines are in the Columbia (Col) ecotype and were grown on soil under a 16 h- light/8-h dark cycle. The *main-2* (GK-728H05), *main-3* (hypomorphic allele), *mail1-1* (GK-840E05), *pp7l-2* (SALK_003071), *morc1-2* (SAIL_893_B06), and *mail1-1 pp7l-2* null mutant lines were previously described ([10], [13], [14], [15], [16], [20], [21]). The *main-2 morc1-2* and *mail1-1 morc1-2* double mutants were obtained by crosses and confirmed by PCR-based genotyping and RT–qPCR analyses. The *pUBQ-MORC1/main-2* and *pUBQ-MORC1/mail1-1* lines were obtained by plant transformation using the *Agrobacterium*-mediated floral dip method ([22]). The two *pUBQ-MORC1/morc1-2* lines #1 and #2 were obtained by crossing *morc1-2* with *pUBQ-MORC1/main* line #2 and *pUBQ-MORC1/mail1* line #1, respectively, followed by subsequent PCR-based genotyping. The complementing lines expressing an epitope-tagged genomic version of PMD or PP7L in corresponding mutant backgrounds were previously described ([16]).

### Cloning of *pUBQ-MORC1*

The pENTR Gateway (GW) vector carrying *MORC1* CDS without STOP codon was obtained from the Jacobsen laboratory. The 3xFLAG tag was subcloned using an AscI site downstream of the cDNA, and DNA integrity was verified by Sanger sequencing (Eurofins). To generate *pUBQ-MORC1*, the *MORC1-3xFLAG* construct was then mobilized into the GW-compatible pUBQ10:GW vector by LR Clonase (Thermo Fisher Scientific) according to the manufacturer's instruction ([18]). *pUBQ-MORC1/main-2* and *pUBQ-MORC1/mail1-1* primary transformants were selected by spraying glufosinate as a selection marker, and resistant plants were saved for further characterization. Primer sequences are described in Table S7.

### Immunoblotting

Total proteins were extracted from leaves of 3-wk-old seedlings using 8 M urea and denatured in Laemmli buffer for 5 min at 95°C. 10–15 µl of protein extracts were run on 10% SDS–PAGE, and proteins were detected by Western blotting using Anti-FLAG M2 monoclonal antibody–peroxidase conjugate (A8592; Sigma-Aldrich) at a dilution of 1:10,000. Western blots were developed using Substrat HRP Immobilon Western (WBKLS0500; Merck Millipore).

### RNA extraction

Total RNA was extracted from leaves of 3-wk-old seedlings grown on soil using Monarch Total RNA Miniprep Kit (T2010; New England Biolabs) according to the manufacturer's protocol.

### RNA sequencing

RNA-seq libraries were generated from 1 µg of input RNA using NEBNext Ultra II Directional RNA Library Prep Kit for Illumina (E7490; New England Biolabs) according to the manufacturer's protocols.

Libraries were sequenced on an Illumina NextSeq 550 machine (Bio-environment platform, UPVD). Reads were trimmed using Trimmomatic ([23]) and mapped to the *A. thaliana* genome (*Arabidopsi*s TAIR10 genome) using HISAT2 ([24]). The sequence alignment files were sorted by name and indexed using SAMtools ([25]). Files were converted to BAM files and a number of reads mapped onto a gene calculated using HTSeq-count ([26]). Differentially expressed genes were obtained with DESeq2 ([27]), using a $\log_2$ fold change $\geq 2$ (up-regulated genes) or $\leq -2$ (down-regulated genes) with an adjusted $P < 0.01$. Principal component analyses were produced using DESeq2 and ggplot2 R packages. Heatmap visualizations were realized using the heatmap2 function from the R gplots package. Boxplots were realized using the boxplot function from R. UpSet plot analyses were performed using the Intervene's UpSet module interface described at https://asntech.shinyapps.io/intervene/ ([28], [29]). RNA sequencing mapping and coverage statistics are described in Table S8.

### RT–qPCR

1 µg of input RNA was converted to cDNA using GoScript Reverse Transcriptase (A501C; Promega) according to the manufacturer's protocol. The final reaction was diluted six times with RNase-free water. RT–qPCR experiments were performed with 4 µl of cDNA combined with the Power Track SYBR Green Master Mix (Thermo Fisher Scientific) using a LightCycler 480 instrument (Roche). Amplification conditions were as follows: 95°C for 5 min; 40 cycles of 95°C for 15 s and 60°C for 1 min; and melting curves. RT–qPCR analyses used the $2^{-\Delta\Delta Ct}$ method. For each analysis, $\Delta Ct$ was first calculated based on the housekeeping *RHIP1* gene Ct value ([30]). $\Delta\Delta Ct$ values were then obtained by subtracting the WT $\Delta Ct$ from the $\Delta Ct$ of each sample. Values were represented on bar charts relative to WT. Three technical replicates were performed per biological replicate, and three biological replicates were used in all experiments. Primer sequences are described in Table S7.

# Data Availability

Nucleotide sequencing data generated in this study have been deposited in European Nucleotide Archive under the accession number PRJEB52795.

# Supplementary Information

# Acknowledgements

The authors thank the Steve Jacobsen laboratory who provided the MORC1 CDS construct as a gift. They also want to acknowledge the LGDP platform members and UPVD Bio-environment facility for their outstanding technical assistance and plant care. This study was supported by "Laboratoires d'Excellence" (LABEX) AGRO (10-LABX-0001)—Agropolis Foundation (project ID 2101-009) and by a UPVD BQR grant. It was set within the framework of the LABEX TULIP (ANR-10-LABX-41) and "Ecole Universitaire de Recherche (EUR)"

TULIP-GS (ANR-18-EURE-0019). L Jarry is a recipient of the PhD fellowship "Bourse région Occitanie 2021." The funders had no role in study design, data collection and analysis, decision to publish, or preparation of the article.

## Author Contributions

L Jarry: conceptualization, resources, data curation, formal analysis, validation, investigation, visualization, methodology, and writing—original draft, review, and editing.

J Descombin: conceptualization, validation, investigation, visualization, methodology, and writing—review and editing.

M Nicolau: conceptualization, validation, investigation, and writing—review and editing.

A Dussutour: validation and investigation.

N Picault: conceptualization, resources, data curation, investigation, visualization, and writing—review and editing.

G Moissiard: conceptualization, supervision, funding acquisition, investigation, visualization, methodology, project administration, and writing—original draft, review, and editing.

## Conflict of Interest Statement

The authors declare that they have no conflict of interest.

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
