## [Reviewer comments · Life Science Alliance]

Plant Mobile Domain proteins ensure *Microrchidia 1* expression to fulfil transposon silencing

Lucas Jarry, Julie Descombin, Melody Nicolau, Nathalie Picault, Ange Dussutour, and Guillaume Moissiard
DOI: <https://doi.org/10.26508/lsa.202201539>

Corresponding author(s): Guillaume Moissiard, Plant Genome and Development Laboratory

Review Timeline:	Submission Date:	2022-05-30
	Editorial Decision:	2022-07-28
	Revision Received:	2022-11-25
	Editorial Decision:	2023-01-03
	Revision Received:	2023-01-10
	Accepted:	2023-01-11

Transaction Report:

July 28, 2022

Re: Life Science Alliance manuscript #LSA-2022-01539-T

Guillaume Moissiard
LGDP UMR5096 CNRS/UPVD
Bâtiment T
58 Avenue Paul Alduy
Perpignan, PO 66860
France

Dear Dr. Moissiard,

Thank you for submitting your manuscript entitled "Plant Mobile Domain proteins are required for Microrchidia 1 expression to fulfil transposon silencing" to Life Science Alliance. The manuscript was assessed by expert reviewers, whose comments are appended to this letter. We invite you to submit a revised manuscript addressing the Reviewer comments.

Thank you for this interesting contribution to Life Science Alliance. We are looking forward to receiving your revised manuscript.

Sincerely,

B. MANUSCRIPT ORGANIZATION AND FORMATTING:

Reviewer #1 (Comments to the Authors (Required)):

In this manuscript, Descombin et al. described that TE silencing mechanism by plant mobile proteins, MAIN and MAIL1 depending Microchidia1 (MORC1) down expression. They found MORC1 was downregulated in pmd mutants and in their interactor PP7L mutant by surveying RNA-seq data. It was carefully shown the expression level of specific target TEs in pmd morc1 double mutant was not significantly higher than in each single mutants. Thus authors concluded that pmd morc1 double mutants do not exacerbate TE silencing defects. Next, authors tested whether the MORC1 complementation by exogenous expression cassette in pmd mutants restore the silencing of target TEs. These findings provide novel information of gene/transposon silencing mechanism to plant science field. However, I have raised several concerns as follows.

Major concerns:

1. Line 144-147, it was observed many misregulated genes in MORC1OE/main because of MORC1 unspecific effect. Thus, I think it is not clear that the expression of target gene/TEs in main was changed by 'MORC1 specific' effect by rescuing MORC1 expression. To reveal MORC1 specific effect, I think following two data set is needed. (1) vector control (pUBQ10::FLAG-OCS). It is needed to show vector transformation itself was not affect TE/gene re-silencing. (2) expression data of different line of MORC1OE/main. There is a possibility that MORC1 unspecific effect may depends on position effect of T-DNA insertion.
2. In figure 3B, several lines of MORC1OE/morc1-2 were shown, however, only one line was analyzed the expression. To conclude as 'MORC1-FLAG expression was sufficient to restore the silencing of several misregulated TEs and DNA-methylated genes in morc1-2 (Line 122-123)', it is needed to show the target expression in several lines.
3. The results indicated that there are two type of category of genes for PMD regulated TEs and genes, MORC1 dependent and independent. Such genes were identified in Figure 4, thus is there any feature difference between two category (e.g. GO term, promoter structure, position of genome, etc) ?

Minor comments:

1. Line 33, the sentence 'TE transposition can have dramatic consequences for the host cell.' is abstract representation. It should be expressed more specific explanation.
2. Line 56, 'largely misunderstood' is better to be replaced as 'largely unclear'.
3. Line 219, Figure S4 is not exist.
4. In Figure legends of Fig1A, it is better to add more precise explanation for Upset plot.

Reviewer #2 (Comments to the Authors (Required)):

In this manuscript the authors analysed the interplay between PMD and MORC1-mediated TE silencing pathways. Loss of function of morc1 and of individual pmd proteins (mail1 and main) was previously shown to lead to release of TE silencing, however, how these proteins regulate TE silencing is still unknown. It was also shown previously that loss of function of pmd proteins leads to down regulation of numerous genes, including MORC1. To find out if this downregulation of MORC1 might explain the TE silencing defects, main morc1 double mutants were generated and RNAseq analyses comparing both single mutants with the double mutant were performed. From this the authors concluded that the main single mutant showed a stronger TE silencing defect than the morc1 mutant and that this effect was not increased in the double mutant and thus that MORC1 might act downstream of MAIN. To test this, MORC1 was ectopically expressed in the main and the mail1 mutant backgrounds. RNAseq analyses from these lines showed that the silencing defect of at least some TEs was rescued indicating that MORC1 downregulation in the pmd background might indeed account for at least some of the silencing defect. The presented analyses seem to be solid. However, the data are lacking any new mechanistic insight into how these silencing pathways might work and it is unclear if the observed rescue of the silencing defects upon MORC1-oe is direct or not. Therefore, they provide only a very limited contribution to the understanding of silencing pathways in plants.

More details below:

- 1) Figure 1B and Figure 2: The transcript level of MORC1 is also shown to be down-regulated in main-3 and main-3/ddc (Why is

this double mutant important here?) and in pp71-2. Does the main-3 mutant and the pp71 mutant also show mis-expression of loci that seem to be regulated downstream of main and mail1 by MORC1 (e.g. those that were tested in Figure 2I-K)?

To prove that MORC1 acts downstream of MAIN and MAIL1 the authors should test if PMD proteins are indeed targeted to the MORC1 promoter.

2) Lines 143: What is the rationale behind using an OE construct instead of a construct in which MORC1 expression is driven by the endogenous MORC1 promoter?

3) Lines 144 to 145: Several genes were found to be mis-expressed in the main/MORC1-oe line and in WT/MORC1-oe and thus independent of the main mutation. What is the evidence that the "rescue" of the TE loci in main/MORC1-oe is not also an unspecific effect that might occur in any silencing defective mutant background?

4) The mail1 morc6 double mutant was shown to exhibit a synergistic effect on TE release. Since MORC1 and MORC6 are known to act in one complex, how do the authors explain this?

5) The authors found that 22 TEs are controlled in main-2 in a MORC1-dependent manner (since they were rescued in the main/MORC1-oe line). Were these TEs also shown to be up-regulated in previously published RNAseq analyses of the mail1 and pp71 mutant background?

Discussion:

Line 213: Did the authors test if PMD proteins bind to chromatin? Why do they suggest that this might occur through another TF? What is Figure S4?

Introduction:

- Line 48-49: it would be helpful to explain in more detail how this mechanism might work.
- The naming of the main mutants is confusing, in the figures it is written main-2 or main-3 while in the text just main.

Reviewer #3 (Comments to the Authors (Required)):

Descombin and colleagues set out to investigate how Plant Mobile Domain (PMD) proteins relate to transposable element (TE) silencing. Upon analysis of RNA-seq of PMD mutants, they discovered that MORC1, a gene previously described to play a role in TE silencing, was among the downregulated genes. They also show that transgenic expression of MORC1 rescues the TE derepression observed in PMD mutants.

While the work shed light into how a fraction of the TEs become upregulated in PMD mutants, the conclusions that can be drawn from the results are very limited.

First, TE expression appears not to have a crucial impact in the phenotype of PMD mutants, as rescue with MORC1 transgene still shows the characteristic developmental problems despite less TE expression (also, MORC1 mutants show normal development).

Second, based on the data it is unclear if PMD proteins play a direct role in TE silencing or not: while transgenic MORC1 expression rescues some TE derepression, it is entirely possible that the other unchanged TEs may be derepressed for similar reasons (e.g. PMDs are required for the expression of yet another gene involved in a bona fide TE repression mechanism). PMD mutants have thousands of downregulated genes and it would be interesting to find out if other key genes involved in TE repression pathways are also not among them. As it stands, the manuscript offers very limited insight into how PMDs impact transcription of MORC1 and or MORC1-independent TEs. I don't see how the authors can learn more about these questions without pursuing lengthy additional experiments and screens, which I would consider inadequate for a reasonable review process.

Considering these limitations, I can offer below a few suggestions to help authors strengthen the manuscript.

1. The results of Figure 1A could be better analyzed. From these common set of 26 genes the authors used to ID MORC1, are there any other category that might be enriched? A GO term enrichment can perhaps be performed?
2. On Figure 2D genotypes/replicates could also be clustered by similarity.
3. The same suggestion made on #1 could be applied to the relevant intersections of differentially expressed genes shown on Figure 2E.
4. It would be particularly interesting to include the MORC1-2 dataset on the Venn diagram of Figure 4B. One would be able to visualize if the same TEs upregulated on MORC1-2 are being rescued in the MORC10E/MAIN-2.
5. Are there any overlapping TF motifs enriched in the promoters of downregulated genes highlighted on Figure 1A that have a match in the promoter of MORC1?
6. The number of upregulated genes in main-2 suggests it impacts or is directly involved in gene repression. Are there any general repressors other than MORC1 among the downregulated genes in main-2? Maybe these could explain some extend of the synergistic effect observed between MORC1 and PMDs?

Reviewer #1 (Comments to the Authors (Required)):

In this manuscript, Descombin et al. described that TE silencing mechanism by plant mobile proteins, MAIN and MAIL1 depending Microchidia1 (MORC1) down expression. They found MORC1 was downregulated in *pmd* mutants and in their interactor PP7L mutant by surveying RNA-seq data. It was carefully shown the expression level of specific target TEs in *pmd morc1* double mutant was not significantly higher than in each single mutants. Thus authors concluded that *pmd morc1* double mutants do not exacerbate TE silencing defects. Next, authors tested whether the MORC1 complementation by exogenous expression cassette in *pmd* mutants restore the silencing of target TEs. These findings provide novel information of gene/transposon silencing mechanism to plant science field. However, I have raised several concerns as follows.

Major concerns:

1. Line 144-147, it was observed many misregulated genes in MORC1OE/main because of MORC1 unspecific effect. Thus, I think it is not clear that the expression of target gene/TEs in main was changed by 'MORC1 specific' effect by rescuing MORC1 expression. To reveal MORC1 specific effect, I think following two data set is needed. (1) vector control (pUBQ10::FLAG-OCS). It is needed to show vector transformation itself was not affect TE/gene re-silencing. (2) expression data of different line of MORC1OE/main. There is a possibility that MORC1 unspecific effect may depends on position effect of T-DNA insertion.

Thank you for this important comment. To refine the subset of loci that can be complemented by MORC1 (MORC1 specific effect) and evaluate MORC1 unspecific effect, we performed RNA-seq analyses using the three other pUBQ-MORC1/*pmd* lines: pUBQ-MORC1/main line 1, pUBQ-MORC1/mail1 line 1 and pUBQ-MORC1/mail1 line 2, and combined them with the previous RNA-seq of pUBQ-MORC1/main line 2. Altogether, these analyses define with a higher accuracy the subset of TEs and DNA-methylated genes for which the silencing is rescued by complementing MORC1 expression in the *pmd* mutants.

Although some variations can be observed between independent lines, those variations are mostly occurring at differentially expressed genes. We explain in the text that they are most likely a sign of MORC1 unspecific effect due to the transgene position effect.

We have also included two additional lines called pUBQ-MORC1/main neg. and pUBQ-MORC1/mail1 neg. that do not accumulate MORC1 protein (although they carry the pUBQ-MORC1 transgene). In these two lines, the TE silencing defects are similar to respective *pmd* mutants. These two lines corroborate our conclusion that MORC1 expression is required for the rescue of TE silencing defects.

2. In figure 3B, several lines of MORC1OE/*morc1-2* were shown, however, only one line was analyzed the expression. To conclude as 'MORC1-FLAG expression was sufficient to restore the silencing of several misregulated TEs and DNA-methylated genes in *morc1-2* (Line 122-123)', it is needed to show the target expression in several lines.

We apologize as former panel 3B was misleading. As mentioned in the text and figure 3 legend of 1st draft, samples #1 to 6 were not from independent lines but from several individuals of the same line (originally named MORC1OE/*morc1-2*, and renamed pUBQ-MORC1/*morc1* line 2 in the revised manuscript). We are now including in panels 3B and 3C a second line called pUBQ-MORC1/*morc1* line 1. As previously mentioned in the materials and methods (but now also stated in the main text), please note that pUBQ-MORC1/*morc1* line 1 derives from the cross between pUBQ-MORC1/*main* line 2 and *morc1*, while pUBQ-MORC1/*morc1* line 2 derives from the cross between pUBQ-MORC1/*mail1* line 1 and *morc1*. This is important because it allows us to compare the effects of same transgene insertion events in the *pmd* and *morc1* backgrounds.

3. The results indicated that there are two type of category of genes for PMD regulated TEs and genes, MORC1 dependent and independent. Such genes were identified in Figure 4, thus is there any feature difference between two category (e.g. GO term, promoter structure, position of genome, etc) ?

We now provide GO term analyses, position in the genome and epigenetic status of misregulated loci in Tables S5 and S6.

Minor comments:

1. Line 33, the sentence 'TE transposition can have dramatic consequences for the host cell.' is abstract representation. It should be expressed more specific explanation.

The sentence has been rewritten precisig that 'TE transposition can disrupt gene sequence'.

2. Line 56, 'largely misunderstood' is better to be replaced as 'largely unclear'.

The sentence has been modified as suggested by the reviewer.

3. Line 219, Figure S4 is not exist.

The original figure S4 was replaced by supplementary "Source Data" files. The misleading references to Figure S4 in the text have been changed to Figure 4G.

4. In Figure legends of Fig1A, it is better to add more precise explanation for Upset plot.

Figure legend has been reformulated to include more details about Upset representation. We also added reference 29 (Lex et al., 2014) in the RNA sequencing section of Materials and Methods.

Reviewer #2 (Comments to the Authors (Required)):

In this manuscript the authors analysed the interplay between PMD and MORC1-mediated TE

silencing pathways. Loss of function of *morc1* and of individual *pmd* proteins (*mail1* and *main*) was previously shown to lead to release of TE silencing, however, how these proteins regulate TE silencing is still unknown. It was also shown previously that loss of function of *pmd* proteins leads to down regulation of numerous genes, including *MORC1*. To find out if this downregulation of *MORC1* might explain the TE silencing defects, *main morc1* double mutants were generated and RNAseq analyses comparing both single mutants with the double mutant were performed. From this the authors concluded that the *main* single mutant showed a stronger TE silencing defect than the *morc1* mutant and that this effect was not increased in the double mutant and thus that *MORC1* might act downstream of *MAIN*. To test this, *MORC1* was ectopically expressed in the *main* and the *mail1* mutant backgrounds. RNAseq analyses from these lines showed that the silencing defect of at least some TEs was rescued indicating that *MORC1* downregulation in the *pmd* background might indeed account for at least some of the silencing defect.

The presented analyses seem to be solid. However, the data are lacking any new mechanistic insight into how these silencing pathways might work and it is unclear if the observed rescue of the silencing defects upon *MORC1*-oe is direct or not. Therefore, they provide only a very limited contribution to the understanding of silencing pathways in plants.

More details below:

1) Figure 1B and Figure 2: The transcript level of *MORC1* is also shown to be down-regulated in *main-3* and *main-3/ddc* (Why is this double mutant important here?) and in *pp7l-2*. Does the *main-3* mutant and the *pp7l* mutant also show mis-expression of loci that seem to be regulated downstream of *main* and *mail1* by *MORC1* (e.g. those that were tested in Figure 2I-K)?

We agree with reviewer 2 that *ddc main-3* does not provide any valuable information, hence its removal. Regarding the upregulated TEs in *main-3* and *pp7l* mutants (described in REF 16), we now provide Venn diagram analyses showing overlaps between *MORC1*-rescued lines and *main-3* or *pp7l*. These analyses revealed that most of complemented TEs in *MORC1*-rescued lines belong to the sets of upregulated TEs in *main-3* and *pp7l* mutants.

To prove that *MORC1* acts downstream of *MAIN* and *MAIL1* the authors should test if *PMD* proteins are indeed targeted to the *MORC1* promoter.

Indeed, we agree with reviewer 2 that it would be nice to show that *PMD* proteins are targeted to the *MORC1* promoter and/or others genes carrying the "DOWN" motif in their promoters. That will be part of another study as we did not have enough time to perform and validate experiments testing this hypothesis.

2) Lines 143: What is the rationale behind using an OE construct instead of a construct in which *MORC1* expression is driven by the endogenous *MORC1* promoter?

As *MORC1* is downregulated in the *pmd* mutants, the rationale is to use an endogenous promoter for which gene expression is not impaired in the mutants. This is now stated in the main text.

3) Lines 144to 145: Several genes were found to be mis-expressed in the *main/MORC1*-oe line and in *WT/MORC1*-oe and thus independent of the *main* mutation. What is the evidence that the "rescue" of the TE loci in *main/MORC1*-oe is not also an unspecific effect that might occur in any silencing defective mutant background?

That is a fair point. Unfortunately, we cannot address this question using our plant material. However, it is very unlikely that *pUBQ-MORC1* would complement an epigenetic mutant such as for instance *morc6* or *mom1* as *MORC1* is not downregulated in these mutants, and *MORC6* or *MOM1* would still be missing to perform their function. On the other hand, we believe that if we had for instance transformed the

pmd mutants with a MORC6 CDS under the control of pUBQ, we would not have seen any complementation of TE silencing defects because MORC6 is not downregulated in the *pmd* mutants.

4) The mail1morc6 double mutant was shown to exhibit a synergistic effect on TE release. Since MORC1 and MORC6 are known to act in one complex, how do the authors explain this?

This is now more precisely discussed in the main text. As previously shown, *morc6* mutant show a stronger TE misregulation than *morc1*, which could explain the discrepancy between *mail1 morc1* and *mail1 morc6*.

5) The authors found that 22 TEs are controlled in main-2 in a MORC1-dependent manner (since they were rescued in the main/MORC1-oe line). Were these TEs also shown to be up-regulated in previously published RNAseq analyses of the mail1 and pp7l mutant background ?

Compelling additional RNA-seq data of pUBQ-MORC1/*pmd* lines and *mail1* mutant allowed us to refined the list of complemented TEs. As mentioned in point 1), most of complemented TEs are upregulated TEs in *mail1* or *pp7l*.

Discussion:

Line 213: Did the authors test if PMD proteins bind to chromatin Why do they suggest that this might occur through another TF? What is Figure S4?

Unfortunately, we could not show PMD/chromatin interaction and we hope that will be part of another study. We propose in the discussion that a TF could be the link between the PMD/PP7L complex and the "DOWN" motif because at the moment, there is no evidence that PMD can directly bind chromatin/DNA.

Introduction:

• **Line 48-49: it would be helpful to explain in more detail how this mechanism might work.**

This is now done. Thank you.

• **The naming of the main mutants is confusing, in the figures it is written main-2 or main-3 while in the text just main.**

This is all fixed in the text.

Reviewer #3 (Comments to the Authors (Required)):

Descombin and colleagues set out to investigate how Plant Mobile Domain (PMD) proteins relate to transposable element (TE) silencing. Upon analysis of RNA-seq of PMD mutants, they discovered that MORC1, a gene previously described to play a role in TE silencing, was among the downregulated genes. They also show that transgenic expression of MORC1 rescues the TE derepression observed in PMD mutants.

While the work shed light into how a fraction of the TEs become upregulated in PMD mutants, the conclusions that can be drawn from the results are very limited.

First, TE expression appears not to have a crucial impact in the phenotype of PMD mutants, as rescue with MORC1 transgene still shows the characteristic developmental problems despite less TE expression (also, MORC1 mutants show normal development).

Second, based on the data it is unclear if PMD proteins play a direct role in TE silencing or not: while transgenic MORC1 expression rescues some TE derepression, it is entirely possible that the other unchanged TEs may be derepressed for similar reasons (e.g. PMDs are required for the expression of yet another gene involved in a bona fide TE repression mechanism). PMD mutants have thousands of downregulated genes and it would be interesting to find out if other key genes involved in TE repression pathways are also not among them. As it stands, the manuscript offers very limited insight into how PMDs impact transcription of MORC1 and or MORC1-independent TEs. I don't see how the authors can learn more about these questions without pursuing lengthy additional experiments and screens, which I would consider inadequate for a reasonable review process.

Considering these limitations, I can offer below a few suggestions to help authors strengthen the manuscript.

1. The results of Figure 1A could be better analyzed. From these common set of 26 genes the authors used to ID MORC1, are there any other category that might be enriched? A GO term enrichment can perhaps be performed?

We did perform a GO term enrichment analysis using AgriGOv2 and amigo, but we could not retrieve any enrichment. This is mentioned in the text.

2. On Figure 2D genotypes/replicates could also be clustered by similarity.

Indeed, that is a good idea. We now provide in panel S1B a Principal component analyses (PCA) showing clustering of biological replicates for all the genetic backgrounds depicted in figure 2D.

3. The same suggestion made on #1 could be applied to the relevant intersections of differentially expressed genes shown on Figure 2E.

This is now provided in Table S3.

4. It would be particularly interesting to include the MORC1-2 dataset on the Venn diagram of Figure 4B. One would be able to visualize if the same TEs upregulated on MORC1-2 are being rescued in the MORC10E/MAIN-2.

That is indeed a very good point. We now show in boxplot and Venn diagram forms the effects of MORC1 complementation in *pmd* lines on TEs that are upregulated in *morc1* mutant.

5. Are there any overlapping TF motifs enriched in the promoters of downregulated genes highlighted on Figure 1A that have a match in the promoter of MORC1?

Indeed, we previously identified a DNA motif called the "DOWN" motif that is enriched in the promoter of commonly downregulated genes in *pmd* and *pp7l* mutants as described in Nicolau et al., 2020 (ref 16). This is now mentioned in the beginning of the result section paragraph "MORC1 is downregulated in *pmd* mutants.

6. The number of upregulated genes in main-2 suggests it impacts or is directly involved in gene repression. Are there any general repressors other than MORC1 among the downregulated genes in main-2? Maye these could explain some extend of the synergistic effect observed between MORC1 and PMDs?

We agree with reviewer 3 that the synergistic effects seen for upregulated genes in main *morc1* mutant could be the consequence of other gene downregulations observed in main single mutant. Indeed, besides MORC1, there are epigenetic/chromatin factors that are downregulated in main or *mail1* mutant such as for instance: *HAC4/AT1G55970* which encodes a putative acetyltransferase protein, or

several TFs like for instance FRF2/AT3G07500 or other TFs. It will be definitely interesting to decipher this aspect in the future. A sentence mentioning the possibility of additional downregulated gene effects has been added in the main text. We thank reviewer 3 for this comment.

January 3, 2023

RE: Life Science Alliance Manuscript #LSA-2022-01539-TR

Dr. Guillaume Moissiard
Plant Genome and Development Laboratory
Bâtiment T
58 Avenue Paul Alduy
Perpignan, PO 66860
France

Dear Dr. Moissiard,

Thank you for submitting your revised manuscript entitled "Plant Mobile Domain proteins ensure Microrchidia 1 expression to fulfil transposon silencing". We would be happy to publish your paper in Life Science Alliance pending final revisions necessary to meet our formatting guidelines.

- please incorporate responses to the Reviewers' remaining comments into your manuscript
- please upload both your main and supplementary figures as single files
- please add the Twitter handle of your host institute/organization as well as your own or/and one of the authors in our system
- No results are coming up for ENA accession number PRJEB52795, please make sure this is publicly accessible at this point

A. FINAL FILES:

B. MANUSCRIPT ORGANIZATION AND FORMATTING:

Sincerely,

Reviewer #1 (Comments to the Authors (Required)):

The manuscript by Jarry et al. presents that TE silencing by plant mobile proteins is partly depending on MORC1 function. Indeed, I agree with the comment of reviewer #3 that the results are limited, however, the analyses and interpretation are convincing and the conclusion that 'PMD proteins ensure MORC1 expression to fulfil transposon silencing' shed light one of the PMD functions. The authors have addressed and answered the points raised in the previous reviews adequately. I only suggest following minor points.

1. In figure 4B, main-2 and mail1-1 should be italicized.
2. In figure 4G, the direct repression symbol from MAIN/MAIL1/PP7L complex to MORC1-dependent TE silencing may not be necessary.

Reviewer #2 (Comments to the Authors (Required)):

In my previous review I raised concerns about the limited mechanistic insight into how these silencing pathways might work (which was also mentioned by Reviewer 3) and about the specificity of the effect of MORC1-oe on pmc-mediated TE silencing (also mentioned by Reviewer 1).

For the first point the authors argued that these experiments cannot be done at the moment and might be published in another manuscript. For the second point the authors included in the revised version RNA-seq analyses of 3 additional independent MORC1-oe lines, one in the main-2 background and two in mail1-1 background and also analysis of two vector control lines. Although these additional data have substantially improved the manuscript, they also show the very limited effect of MORC1-oe on pmc-mediated silencing. Although the authors claim that "Rescuing MORC1 expression in pmc mutants is sufficient to restore the silencing of a large fraction of TEs" (discussion) it seems, that as shown in Figure 4 only 9 out of 101 TEs are rescued in both main-2/MORC1oe lines and 10 out of 53 in the mail1-1 background lines. And the overlap between rescued TEs in main-2 and mail1-1 background seems to be even lower (this potentially also interesting point is not shown but can be seen from the list in tableS5). As the authors point out in the text there is also quite a variation between independent lines and this further supports the concern that the observed rescue of TE silencing might be an unspecific effect. Based on this and in the absence of any further hint to how pmc genes might regulate MORC1 expression or what distinguishes the loci that are commonly rescued by MORC1-oe in both main and mail1 mutant background from all other mis-expressed loci I am still not convinced that the presented data do provide any significant advance to the scientific community on this topic.

Reviewer #3 (Comments to the Authors (Required)):

Descombin and colleagues delivered a revised version of the manuscript addressing most of the reviewer's comments. I am

satisfied with the improved version and support it for publication. I do however maintain my view that the study offers limited view into the relationship between MAIN/MAIL1 and transposon repression. An experiment to show a direct versus indirect consequence has not been provided (e.g. see reviewer #2 main point 1, second item where the authors reply "...That will be part of another study as we did not have enough time to perform and validate experiments testing this hypothesis.").

In my view of the presented data, an opposite conclusion to the title claim of the manuscript is that it is as equally possible that MAIN/MAIL1 are involved in the transcription of thousands of genes, some of which happen to be important for transposon repression like MORC3. In any case it will be interesting to follow up to see if the author's interpretation is correct.

-please incorporate responses to the Reviewers' remaining comments into your manuscript

done

-please upload both your main and supplementary figures as single files

ok

-please add the Twitter handle of your host institute/organization as well as your own or/and one of the authors in our system

ok

-No results are coming up for ENA accession number PRJEB52795, please make sure this is publicly accessible at this point

We have made the data publically available. It should be reflected on the ENA Browser in a few days.

RELEASE_DATE | STUDY_ID | STUDY_TITLE

06-Jan-2023 | PRJEB52795 (ERP137538) | Complex interplay between Plant Mobile Domain and Microrchidia silencing pathways

Yes, we are planning a press release through CNRS INSB communication office.

Yes Lucas Jarry, 1st author of the manuscript, will submit a video in the coming days.

The manuscript by Jarry et al. presents that TE silencing by plant mobile proteins is partly depending on MORC1 function.

Indeed, I agree with the comment of reviewer #3 that the results are limited, however, the analyses and interpretation are convincing and the conclusion that 'PMD proteins ensure MORC1 expression to fulfil transposon silencing' shed light on one of the PMD functions. The authors have addressed and answered the points raised in the previous reviews adequately.

I only suggest following minor points.

1. In figure 4B, main-2 and mail1-1 should be italicized.

Thank you, typo has been modified.

2. In figure 4G, the direct repression symbol from MAIN/MAIL1/PP7L complex to MORC1-dependent TE silencing may not be necessary.

We agree and made appropriate modification.

Reviewer #2 (Comments to the Authors (Required)):

In my previous review I raised concerns about the limited mechanistic insight into how these silencing pathways might work (which was also mentioned by Reviewer 3) and about the specificity of the effect of MORC1-oe on pmc-mediated TE silencing (also mentioned by Reviewer 1).

For the first point the authors argued that these experiments cannot be done at the moment and might be published in another manuscript. For the second point the authors included in the revised version RNA-seq analyses of 3 additional independent MORC1-oe lines, one in the main-2 background and two in mail1-1 background and also analysis of two vector control lines. Although these additional data have substantially improved the manuscript, they also show the very limited effect of MORC1-oe on pmc-mediated silencing. Although the authors claim that "Rescuing MORC1 expression in pmc mutants is sufficient to restore the silencing of a large fraction of TEs" (discussion) it seems, that as shown in Figure 4 only 9 out of 101 TEs are rescued in both main-2/MORC1oe lines and 10 out of 53 in the mail1-1 background lines. And the overlap between rescued TEs in main-2 and mail1-1 background seems to be even lower (this potentially also interesting point is not shown but can be seen from the list in tableS5). As the authors point out in the text there is also quite a variation between independent lines and this further supports the concern that the observed rescue of TE silencing might be an unspecific effect. Based on this and in the absence of any further hint to how pmc genes might regulate MORC1 expression or what distinguishes the loci that are commonly rescued by MORC1-oe in both main and mail1 mutant background from all other mis-expressed loci I am still not convinced that the presented data do provide any significant advance to the scientific community on this topic.

Thank you for your comments. We have modified the discussion title as followed:

Line 229: 'Rescuing MORC1 expression in pmc mutants is sufficient to restore the silencing of a **subset** of TEs'

We show that the effect on TE silencing is milder in *morc1* loss of function mutant than in *pmc* mutants. Therefore, it makes sense to us that the fraction of TEs belonging to the MORC1-dependent TE silencing class is smaller than the one belonging to the MORC1-independent class. Nevertheless, MORC1-dependent TEs are consistently repressed by rescuing MORC1 expression in the four pUBQ10::MORC1 lines, even in the pUBQ10::MORC1 mail1 line2 for which the box plot statistics are not significant (although the median value is slightly decreased in comparison to mail1 control). Altogether, we think these data support well the statement that MORC1 expression requires the PMD proteins (directly or indirectly to be determined) to fulfil MORC1-mediated TE silencing. Importantly, and as mentioned by reviewer 2, it will be necessary to test if MORC1 expression directly or indirectly requires PMD activity at MORC1 genomic location. Also, as mentioned by reviewer 3, it will be important to identify the molecular components required for the silencing of MORC1-independent TEs.

To take all of these aspects into account, we have slightly modified the last part of abstract, as well as the end of discussion/conclusion and Figure 4G legend.

Reviewer #3 (Comments to the Authors (Required)):

Descombin and colleagues delivered a revised version of the manuscript addressing most of the reviewer's comments. I am satisfied with the improved version and support it for publication. I do however maintain my view that the study offers limited view into the relationship between MAIN/MAIL1 and transposon repression. An experiment to show a direct versus indirect consequence has not been provided (e.g. see reviewer #2 main point 1, second item where the authors reply "...That will be part of another study as we did not have enough time to perform and validate experiments testing this hypothesis.").

In my view of the presented data, an opposite conclusion to the title claim of the manuscript is that it is as equally possible that MAIN/MAIL1 are involved in the transcription of thousands of genes, some of which happen to be important for transposon repression like MORC3. In any case it will be interesting to follow up to see if the author's interpretation is correct.

Thank you for your comments. We agree with them and as mentioned to reviewer 2, we have slightly modified the last part of abstract, as well as the end of discussion/conclusion and Figure 4G legend.

January 11, 2023

RE: Life Science Alliance Manuscript #LSA-2022-01539-TRR

Dr. Guillaume Moissiard
Plant Genome and Development Laboratory
Bâtiment T
58 Avenue Paul Alduy
Perpignan, PO 66860
France

Dear Dr. Moissiard,

Thank you for submitting your Research Article entitled "Plant Mobile Domain proteins ensure Microrchidia 1 expression to fulfil transposon silencing". It is a pleasure to let you know that your manuscript is now accepted for publication in Life Science Alliance. Congratulations on this interesting work.

DISTRIBUTION OF MATERIALS:

Again, congratulations on a very nice paper. I hope you found the review process to be constructive and are pleased with how the manuscript was handled editorially. We look forward to future exciting submissions from your lab.

Sincerely,
